

# Development of a cold atomic muonium beam
# for next generation atomic physics and gravity experiments

Anna Sótér[1*] and Andreas Knecht[2]

**1** ETH Zürich, John-von-Neumann-Weg 9, 8093 Zürich, Switzerland
**2** Paul Scherrer Institut, CH-5232 Villigen, Switzerland

⋆ anna.soter@psi.ch

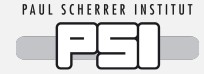

## Abstract

**A high-intensity, low-emittance atomic muonium (M $= \mu^+ + e^-$) beam is being developed, which would enable improving the precision of M spectroscopy measurements, and may allow a direct observation of the M gravitational interaction. Measuring the free fall of M atoms would be the first test of the weak equivalence principle using elementary antimatter ($\mu^+$) and a purely leptonic system. Such an experiment relies on the high intensity, continuous muon beams available at the Paul Scherrer Institute (PSI, Switzerland), and a proposed novel M source. In this paper, the theoretical motivation and principles of this experiment are described.**

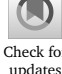
## 31.1 Introduction

Muonium (M) is a two-body exotic atom consisting of a positive anti-muon ($\mu^+$) and an electron ($e^-$). This purely leptonic system can be a unique precision probe to test bound-state QED without the influence of nuclear- and finite size effects. Laser spectroscopy of the M 1S-2S transition [1, 2], and microwave spectroscopy of the M ground state hyperfine structure [3] provided precision measurements of fundamental constants (muon mass, magnetic moment), while searches for muonium-antimuonium conversion put limits on the strength of charged lepton number violation [4]. Improvements in these measurements especially 1S-2S spectroscopy is strongly motivated by recent experiments measuring the anomalous muon $g−2$ [5]. A high intensity, cold atomic beam could significantly improve statistical limitations and systematic effects originating from the (residual) Doppler shift.

Another unique and so far unexplored facet of M is that its mass is dominated by the $\mu^+$, which is not only an elementary antiparticle, but also a second-generation lepton. Direct measurement of the gravitational interaction, thereby tests the weak equivalence principle of such particles, has not yet been attempted [6, 7]. Besides muonium, only antihydrogen

($\bar{\mathrm{H}} = \bar{p} + e^+$) [8–10] and positronium (Ps $= e^- + e^+$) [11–13] have been proposed as laboratory candidates for antimatter gravity experiments, and M is the only viable candidate for testing gravity with purely leptonic, second generation matter.

### 31.1.1 The weak equivalence principle

The Standard Model (SM), as any local, Lorentz-invariant quantum field theory, incorporates CPT symmetry - the simultaneous transformations of charge conjugation (C) parity transformation (P) and time reversal (T) - as an exact symmetry [14]. An important consequence of this is the equivalence of various measurable properties of matter and antimatter, such as the mass, the magnitude of the charge, and the strength of certain interactions. Comparative measurements between matter and antimatter put stringent limits on CPT violation by different experiments using mesons ($K_0 - \bar{K}_0$) [15] leptons ($e^+ - e^-$, $\mu^+ - \mu^-$) [16,17] and baryons ($p - \bar{p}$) [18–21].

With the lack of a unified theory of General Relativity (GR) and the SM, the considerations above however do not imply anything about the gravitational interaction of matter and antimatter. Our expectations originate from the assumed equivalency of the inertial and gravitational masses of particles, which is incorporated in GR as part of the equivalence principle [22,23]. The exact formulation of this principle varies in the literature, and frequently cited as a collective of some these statements below:

1. Weak equivalence principle (WEP) or *universality of free-fall*: all particles (and antiparticles) fall with the same acceleration in a gravitational field.

2. Local position invariance (LPI): The outcome of any local non-gravitational experiment is independent of its location in space or time. Experimental consequences:

   (a) the *universality of clocks* (WEP-c), meaning all systems regardless of their composition (e.g. matter or antimatter) experience the same local time.

   (b) the lack of *variation of fundamental constants* (WEP-v) in time.

3. Local Lorentz invariance (LLI): The outcome of any local non-gravitational experiment in a free-falling laboratory is independent of its velocity.

4. Strong equivalence principle (SEP): states LLI and LPI combined and extended to the gravitational measurements as well (e.g. test bodies with significant contributions from their own gravitational field.)

The combination of the above weak statements (LLI with LPI, sometimes WEP included) is frequently referred to as Einstein's equivalence principle. Most importantly, violation of one of these principles would not necessarily mean the violation of all, and depending on the underlying new physics, it would effect GR and the SM on different levels [23, 24]. Hence, testing the above equivalence principles independently in different experiments using different SM particles is essential [22, 23, 25].

For example, in Earth-based or satellite-borne laboratories, gravitational redshift experiments (WEP-c) and direct free-fall experiments (WEP) using different types of matter may be considered. WEP-c was tested to relatively high accuracy ($\Delta g/g < 10^{-6}$) using matter and antimatter clocks, H and $\bar{\mathrm{H}}$ [18, 24] as well as by measuring cyclotron frequencies of trapped $p$ and $\bar{p}$ [19]. Such experiments arguably also constrain direct WEP-violation originating from certain SM extensions [24,26]. However, direct gravitational free-fall experiments (tests of the WEP) have never been carried out using anything other than normal matter, more precisely macroscopic objects of different material composition, neutral atoms or neutrons.

## 31.2 Experiments for testing the WEP

The most rigorous tests of the WEP utilize Earth-based and satellite-borne experiments that either use the modern versions of the Eötvös torsion pendulum, or other sensitive accelerometers. These experiments compare gravitational accelerations of two macroscopic test masses $(g_1, g_2)$ in terms of the Eötvös parameter

$$\eta(1,2) = 2\frac{|g_1 - g_2|}{|g_1 + g_2|}.\tag{31.1}$$

The highest precision comes from the satellite-borne MICROSCOPE experiment [27] for titanium and platinum, giving $\eta(\text{Ti}, \text{Pt}) = [1 \pm 9(\text{stat}) \pm 9(\text{syst})] \times 10^{-15}$, which is about an order or magnitude better than the best torsion pendulum results from the Eöt-Wash group [28]. On the largest mass scales, the Lunar Ranging Test is the most notable, constraining differences between the Earth and Moon gravitational and inertial mass ratios to levels below $\sim 10^{-13}$ [29].

The WEP has been tested on the atomic scales as well. The latest atom interferometry results comparing two isotopes of rubidium in free-falling cold atom clouds confirmed a null measurement with $\eta(^{85}\text{Rb}, ^{87}\text{Rb}) = [1.6 \pm 1.8(\text{stat}) \pm 3.4(\text{syst})] \times 10^{-12}$ [30].

Gravitational acceleration has only been observed with one subatomic particle, the neutron. The most precise experiments were carried out using neutron refractometers [31], neutron spin-echo technique [32] and also the gravitational quantum states of ultracold neutrons [33, 34]: they have reached an overall precision of $\sim$0.3 %. New experiments plan to improve this by at least an order of magnitude [35].

In summary, WEP tests have limited the Eötvös parameter to $\eta < 1.3 \times 10^{-14}$ for different (macroscopic) elements. Future satellite-borne experiments may improve the precision by two orders of magnitude [23, 36].

### 31.2.1 Possibilities for new physics violating WEP in exotic atoms

Conservative extensions of the SM and GR that would differentiate matter and antimatter in a free fall experiment were discussed with the specific case of antihydrogen [24]. The possibilities discussed include extensions of the existing theories like Kostelecký's extension of the SM [37] containing Lorentz- and CPT violating terms, or minimal modifications of GR that would maintain core principles (like local Lorentz invariance, causality, description as a Riemannian manifold) but modify the dynamics described by the action by adding extra terms that modify the energy-momentum tensor. Several possibilities of 'fifth force' scenarios have also been discussed in the literature, most recently in e.g. Refs. [38, 39]. In Ref. [24] it is pointed out how such new vector bosons could have different couplings to the oppositely charged matter and antimatter, and how this would impact WEP measurements.

The resulting theoretical possibilities are narrow, especially in light of existing WEP measurements on ordinary matter that arguably constrain effects of antimatter gravity via the core principles above and the potential and kinetic energies incorporated in the rest mass [26], and WEP-c measurements that already set constraints on GR extensions [24]. The overall conclusion from theory is that while possible violations of WEP in antihydrogen free-fall experiments may be envisaged, present viable models that do not break the principles of the GR or SM suggest that they are small, and almost certainly already constrained with WEP-c experiments at the $\Delta g/g < 10^{-6}$ level [24]. This consideration also applies to the proposed positronium experiments [11–13] that would probe the antimatter counterpart of the electron.

The same considerations however do not necessarily apply to muonium, which contains an elementary antiparticle from the second generation $(\mu^+)$. Direct gravitational tests have never been carried out before neither with $\mu^+$ nor $\mu^-$. Hence, we may not need to envision long-range vector bosons (fifth forces) that differentiate matter and antimatter to explain an

unexpected result, but could explore other new physics that couples differently to muons than electrons. In the light of recent precision experiments that show intriguing discrepancies in the charged lepton sector like the muon g-2 anomaly [5] or the B anomalies [40], such exotic BSM physics may not be so far fetched.

As to WEP-c tests, next generation experiments of the 1S-2S transition frequency of M have the capability of reaching $\sim 0.1$ ppm fractional precision, and of being sensitive to the effects of gravitational redshift change while the laboratory travels in the solar system (annual modulations of the gravitational potential in perihelion-aphelion) [41]. The interpretation of the muon g-2 result as a clock measurement [5, 41] may also bring some intriguing hints in the same direction.

We also note that there has been an ambiguity in interpreting what experiments with composite objects like neutrons or neutral atoms already tell us about the connection of gravity to the SM particles and interactions [26, 41]. About 99 % of the rest mass of protons and neutrons comes from the strong interaction that confines the constituent quarks. Nuclear binding- and kinetic energies further shift the mass up to $\sim 9$ MeV/c$^2$ per nucleon, while electrostatic interactions with another few eV/c$^2$. In this sense, direct gravity experiments have so far tested mainly binding energies from the strong interaction.

However, the mass of the muonium is dominated by the elementary muon mass, which is a fundamental parameter in the SM. Hence measuring muonium gravity may provide cleaner access to understanding the connection of gravity to elementary particles in the absence of an overwhelming strong interaction.

## 31.3 Prospects for a gravity experiment with a novel M beam

A direct gravity experiment using muonium is inherently challenging due to the short lifetime ($\tau \sim 2.2\ \mu s$) of the $\mu^+$ and the fact that M atoms must be created in matter, while experiments must be carried out *in vacuo*. These imply that we need to envision experiments using propagating atomic beams. A straightforward method is to use atom interferometry, which is known to be a sensitive method to observe inertial forces [30]. However, this requires ultracold atomic clouds, or well-collimated atomic beams with small transverse momentum.

Present vacuum muonium sources are room temperature, porous materials that allow combination of the muon with an electron from the bulk, and a following quick diffusion inside the nanoscopic pores (See Figure 31.1 A). Laser ablated silica aerogel is one of the best room temperature converters; the microscopic holes created by the laser enhance the emission of the M atoms into vacuum. Such sources provide $\sim 3\%$ muon-to-vacuum M conversion using surface $\mu^+$ beams of 28 MeV/c momentum [42]. However, such converters produce a M beam with broad (thermal) energy and angular ($\sim\cos\theta$) distributions.

Mesoporous materials have been shown to convert $\mu^+$ to vacuum M with efficiencies of 40% at room temperature when using a highly moderated, keV energy $\mu^+$ beam; this has an intensity four orders-of-magnitude lower than a surface muon beam. These low-energy muons penetrate only a few $\mu$m into the surface, but are emitted with wide energy- and angular distributions [43]. Improving the source quality by cooling these samples results in lower emission rates, with no observable emission below $\sim 50$ K due to the decreased diffusion constant, and the sticking of M to the pore walls that occurs unavoidably with any conventional M converter [43, 44].

### 31.3.1 Vacuum muonium from superfluid helium

Superfluid helium (SFHe) may overcome the above mentioned difficulties due to its inert nature that rejects impurities from its bulk even at the lowest temperatures. This can be qualitatively explained by the unusually small mean distance ($\sim 0.3$ nm) of the condensed He atoms:

Figure 31.1: (a) Principle of a conventional $\mu^+$-to-vacuum-M converter based on porous materials. (b) Principle of a SFHe-based converter. (c) Comparison of the expected M velocity distribution from SFHe (blue) and a mesoporous (red) converters.

when implanting large impurity atoms or negative ions, nearby He atoms will be repelled by the Pauli core repulsion [45], resulting in a spherical cavity (bubble) around the impurity. This exercises an inward pressure that results in a positive chemical potential of M, that results in the ejection of the impurity from the bulk when they reach the surface.

The principle of the proposed M source relying on this mechanism [6,46] is summarized in Figure 31.1 (b). The $\mu^+$ are stopped in the bulk of SFHe, where they capture an electron from the ionization trails. The M atom formed in the bubble state (M*) diffuses to the surface where it will be emitted perpendicularly, with kinetic energy defined by the chemical potential, only slightly broadened by thermal energies (Figure 31.1 (c)).

The chemical potentials for $^4$He, $^3$He, H, D and T in SFHe have been calculated [47,48], and these predictions have been experimentally verified for $^4$He, $^3$He and D [49]. Modelling M atoms as a light hydrogen isotope gives an approximate chemical potential of $E/k_B \approx 270$ K [50], implying that the M atom will leave the SFHe surface with a well defined longitudinal velocity of $v_M \sim 6300$ m/s. The velocity spread and the transverse velocities are given in first approximation by the thermal motion of the M* bubble in the liquid. Predicting this is difficult without a microscopic theory of the quantum liquid.

Based on [47], the M* acquires an effective mass of $m_M^* \approx 2.5\, m_{He}$ due to hydrodynamic back-flow effects in SFHe, similar to all hydrogen isotopes [50]. In a simplified model, the M* loses energy in a 200 mK isotopically-pure superfluid $^4$He solely by creating rotons and phonons (no scattering on $^3$He), until its kinetic energy falls below the roton gap [51] ($\Delta_{rot}/k_B = 8.6$ K), resulting in thermal velocities distributed below $v_t \approx 110$ m/s. Thermally available phonons are sparse at this temperature, hence scattering on phonons is unlikely on the relevant $\mu$s timescales [52]. The small effective mass of the M* suggests we can neglect other hydrodynamic effects like vortex nucleation as well [53], and assume that M* moves afterwards ballistically in the SFHe medium, with average velocities of $\bar{v}_t \approx v_t/2$. This allows a large fraction of the atoms to escape from $\sim 100$ $\mu$m thick SFHe layers, a thickness that can efficiently stop $\mu^+$ beams of 10-12 MeV/c momentum.

In summary, with the assumptions above and neglecting further surface effects, we expect efficient muon-to-vacuum-M ($\sim 10-30\%$) conversion with a mean atomic velocity of $v_M \approx 6.3$ mm/$\mu$s in the longitudinal direction (originating from the chemical potential), and a spread given approximately as $v_t \approx 0.11$ mm/$\mu$s from the thermal velocities above. This yields to a momentum bite of $< 0.01\%$, and $\alpha \approx v_t/v_M \approx 17$ mrad angular distribution. Moreover, the cold temperature of the SFHe ($\sim 200$ mK) leads to a to a small saturated vapor density (equivalent to UHV conditions at room temperature) which is needed to reduce the collision of the vacuum M with the He gas that would degrade the quality of the M beam.

We have constructed a 200 mK cryogenic target cooled by a dilution refrigerator for the first proof-of-principle experiments to test the above theoretical assumptions, and presently carrying out the first measurements at PSI [54].

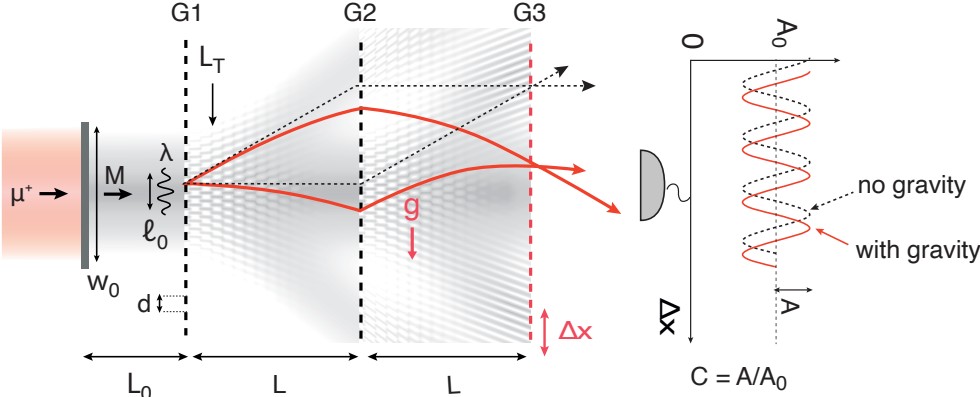

Figure 31.2: A three-grating interferometer used to measure the gravitational inter-action of M atoms. The quantum diffraction pattern caused by the gratings G1 and G2 with a fully coherent beam is given in grey. Classical trajectories (red and dashed lines) are shown to illustrate the effect of gravity on the measured interference pattern appearing at G3. The vertical shift of the interference pattern caused by the gravitational acceleration $g$ is detected by measuring the transmitted M rate while scanning G3 in vertical direction. See details in text.

### 31.3.2 Free fall experiment using M-atom interferometry

If the M atoms are initially at rest in the vertical direction and obey the weak equivalence principle, they fall a mere $\Delta x = \frac{1}{2}gt^2 = 600$ pm in a time of $t = 5\tau$. The measurement of this tiny gravitational fall needs precise knowledge of the initial momentum of the atoms, and requires strict momentum selection. Two periodic gratings (G1 and G2) with horizontal slits of pitch $d$ and spaced by a distance $L$ could be used to achieve this momentum selection as shown in Figure 31.2.

The classical and quantum regime of this device is characterized by de Broglie wavelength of the atoms, $\lambda = h/p$, and grating pitch $d$ in terms of the Talbot length, $L_T = d^2/\lambda$, which is approximately 18 microns for thermal M atoms with $\lambda_M \approx 0.56$ nm. If the grating distances are much smaller than the Talbot length ($L \ll L_T$, the diffraction of the atoms can be neglected during propagation in the device, and this classical device is called a Moiré deflectometer. With the choice of much smaller grating pitch or larger distances $L \gg L_T$ diffraction and in general the wave nature of the atoms become significant, and we work on an interferometer.

With both classical and quantum cases, trajectory selection at G1 and G2 will result in an intensity pattern with the same periodicity $d$ at a distance $L$ after G2. Gravitational acceleration and deflection of the atoms causes a phase shift $\delta\phi$ of this pattern in the vertical direction as $\delta\phi = 2\pi g T^2/d$, where $T = L/v_M$ is the M time of flight between each pair of gratings.

Direct observation of this sub-micron patters and sub-nanometer shifts needed for measuring M gravity would be extremely hard. It is possible however to carry out an indirect measurement using a third grating (G3) of the same pitch $d$, placed at distance $L$ from G2. By counting the total rate of M atoms transmitted through G3 as a function of the G3 vertical position $\Delta x$ the phase shift can be measured.

The contrast of the intensity pattern $C$ is defined by the ratio of the amplitude and the average yield $C = A/A_0$ as shown in Figure 31.2. When the three gratings work as an interferometer, this contrast strongly depends on the transverse coherence length of the beam, $\ell_0$, that determines how many slits of G1 are illuminated with a coherent wavefront. This coherence length in relation to the beam width $w_0$ and the interferometer parameters (the grating

periodicity $d$ and distances $L$) together with the de Broglie wavelength ($\lambda$) of the atoms is sufficient to estimate to describe the interferometer performance in the first approximation. In analogy to statistical optics (Van Cittert-Zernike theorem [55]), we can relate the transverse coherence length of the M beam to the transverse momentum distribution of the atoms: $\ell_0 = \frac{1}{2}\frac{\lambda}{\alpha} \approx 16$ nm, where $\alpha$ is the above mentioned angular spread of the M source. This initial transverse coherence is naturally increasing as the atoms experience diffraction on the first grating. In simplified terms, diffraction results in a new coherent wavefront, that expands along the angle of diffraction. Regardless whether the 3-grating device works in the classical regime or as an interferometer, the sensitivity in measuring the gravitational acceleration $g$ is given by [56]

$$\Delta g = \frac{1}{2\pi T^2}\frac{d}{C\sqrt{N}}\,, \tag{31.2}$$

where $N$ is the number of M atoms transmitted through G3 and measured by the detector given by

$$N = N_0\,\varepsilon_0\,e^{-(t_0+2T)/\tau}\,(T_G)^3\,\varepsilon_{\det}\,, \tag{31.3}$$

with $N_0$ being the number of M atoms produced at the M source, and $\varepsilon_0$ the M transport efficiency from the source to G1. The M decay is accounted for by the third term $e^{-(t_0+2T)/\tau}$, where $t_0$ is the time of flight from the source to G1. The number of detected M atoms is further reduced by the M detection efficiency $\varepsilon_{\det}$, and by the limited transmission $T_G$ of a single grating. The short lifetime of the muon necessitates a gain in sensitivity by using a small grating pitch $d$. Maximal sensitivity, as a tradeoff between phase shift $\delta\phi$ and statistics $N$, is obtained for $T \approx 6-8\,\mu$s corresponding to an interferometer length of 40-50 mm.

A calculation of the interferometer parameters to extract the contrast $C$, uses an approximation of the M source with a Gaussian Schell-model beam [57], and adapted mutual intensity functions that are widely used to describe the propagation of partially coherent light [55]. Using realistic parameters on the initial beam size and quality expected from the superfluid source above, the fringe contrast of $C \approx 0.3$ at the exact position of G3 can be achieved. The contrast in this three-grating setup is less sensitive to the beam quality, but the sensitivity of the high contrast region along the propagation axis is, and shrinks to few $\mu$m. Such a measurement thus requires precise G3 positioning with $\mu$m-accuracy in the optical axis, and below-nm-accuracy in the vertical direction.

From (31.2) we see that determining the sign of $g$ (more precisely to reach $\Delta g/g = 1$) in about one day, requires the detection of 3.2 M/s, assuming a contrast $C = 0.3$. Following (31.3), and taking pessimistic estimates from Monte Carlo simulations and initial detector and grating studies studies by using $T_G = 0.3$, $\varepsilon_0 = 0.75$ and $\varepsilon_{\det} = 0.3$, at the source we need $N_0 \approx 1.4 \times 10^4$ M/s. As a comparison the $\pi$E5 beam line at PSI can presently deliver $3.6\times10^6\,\mu^+$/s at a momentum of 10 MeV/c within a transverse area of about 400 mm$^2$. At this muon momentum we can expect a muon-to-vacuum-M conversion efficiency of about 0.1-0.3 based on the above discussion. This will result in M rates of up to $\sim 1.1 \times 10^6$ M/s. These high rates may allow a further collimation of the M beam to a $5 \times 1$ mm area, which would put less strain on grating production and alignment and would cut the number of useful M atoms conservatively by a factor 5 mm$^2$/400 mm$^2$ = 0.013. Using these parameters where there is room for contingency, we expect to produce the necessary rate of $\sim 5 \times 10^4$ M/s in an small area of $\sim 5 \times 1$ mm$^2$, and reach the goal sensitivity of $\Delta g = \frac{9.8 \text{ m/s}^2}{\sqrt{\# \text{ days}}}$ with present $\mu^+$ sources.

An increase by two orders of magnitude in $\mu^+$ rates expected by the proposed HIMB (High Intensity Muon Beam [58]) project at PSI will further improve the sensitivity of to $g$.

## 31.4 Summary and outlook

With the development of a novel, cold atomic M beam with high yields of $10^4-10^5$ M/s and angular divergence of $\alpha \sim 10-20$ mrad, direct measurement of the gravitational acceleration of M seems feasible on a $\Delta g/g = 10^{-2}$ level of precision. While this precision is not comparable to present tests of the equivalence principle using normal matter ($\Delta g/g < 10^{-15}$), this experiment would be the first direct free fall demonstration using second generation (anti)matter. Moreover, the purely leptonic content of the atom would make it possible to study gravity for the first time in the absence of large binding energies from the strong interaction.

We are presently carrying out feasibility studies, and developing the first prototype of the cryogenic atomic source and the accompanying detector system needed for this experiment at PSI. We are also investigating further theoretical aspects using realistic M beams, and working on production methods for the 100-nm-pitch M interferometer and stabilization methods needed for this precision.

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
