# Peer review of "Development of a cold atomic muonium beam for next generation atomic physics and gravity experiments"

_SciPost Physics Proceedings, doi:SciPost Phys. Proc. 5, 031 (2021)_

## Round 1 · Referee Report · Lars Gerchow (Referee 1) · 2021-7-3

Report

In the manuscript scipost_202106_00044v1, with the title „Development of a cold atomic muonium beam for next generation atomic physics and gravity experiments“, by A. Soter and A. Knecht, the authors present the theoretical motivation and a concept to study the interaction of muonium (M) with gravity. While the idea of using free fall M as a test of the Weak Equivalence Principle (WEP) is not new, it has never been explored experimentally and is highly interesting. With M being purely leptonic such a measurement would be truly complementary to current measurements of the WEP.
The theoretical background and motivation to perform such an experiment is incredibly well presented. The manuscript as a whole is very well written and suited for the publication SciPost Special Volume 'Particle Physics at PSI'.
However, there are some remarks that I would like to address to the authors.
Therefore, in my opinion, it should be published in SciPost after minor revision.

Requested changes

1 - The authors name the existence of 'fifth force' theories in the literature (line 105f). It would be appreciated if the corresponding publications describing these are cited for the interested reader. 2- The literature presenting the constrains on ∆g/g from WEP-c experiments and the mentioned to be 'proposed positronium experiments' should also be cited accordingly. (line 115f) 3- In the abstract the concepts of the cold M source is claimed to enable improved M spectroscopy. Besides the mention of improvements through increased statistics and less systematics (line 27f) this topic is not further addressed. It would be great to e.g. slightly expand the M & WEP-c discussion in line 125ff. I.e. what is the current level, which are the mentioned next generation experiments and how would the M source presented in this manuscript be able to improve on this. (If even, coupling the necessary laser into the SFHe environment seems to be quite a challenge on first thoughts) 4- The authors present and justify the use of SFHe very well in 31.3.1. However, when coming to estimate the muon-to-vacuum conversion (line 195f) are quick to neglect potential surface effects without much justification. Is this a learning from the experiments mentioned in line 178? Why cant M 'stick' to SFHe with some surface states similar to the 'conventional converters'? 5- In the presentation of the concept of the three-grating interferometer 31.3.2 many different quantities and values are quoted and justified in the text. It would be great to have them presented in a more concise way to conclude on the many aspects necessary to do this experiment, e.g. by introducing the values in Fig. 31.2. 6- The authors explain that the positioning of the grating G3 has to fulfill a certain translation accuracy (line 254f). While it is clearly linked, what about the accuracy needed in the angular alignment of the gratings? Moreover, even if much less stringent, the positioning of G1 and G2 will also have its constraints? 7- When the authors conclude on the signal rate and sensitivity of the proposed experiment most values used in the calculation are well justified. However, the detection efficiency is simply assumed to be 30% without any explanation (line 258). As this is a very critical parameter and the detection in such an environment is anything but easy the reviewer would appreciate if more details on how this value is derived would be given. 8- The reviewer would like to ask the authors to also use the abbreviation for HIMB (line 268) an provide an adequate citation. Moreover, an increased rate almost always lead to an increased sensitivity. If possible, could the authors state a quantitative estimation? 9- The implementation of hyperlinks in the references should be done for all items. The reviewer would also like to ask the authors to perform this practice prior to their future submissions as it enables a reviewer to conduct a better review through spending more time on what matters.

  • validity: -
  • significance: -
  • originality: -
  • clarity: -
  • formatting: -
  • grammar: -

Author:  Anna Soter  on 2021-07-31  [id 1628]

(in reply to Report 1 by Lars Gerchow on 2021-07-03)

Thank you very much for the detailed and thorough comments.

1 - The authors name the existence of 'fifth force' theories in the literature (line 105f). It would be appreciated if the corresponding publications describing these are cited for the interested reader.

--> We cited some of the corresponding theories, and re-cited here the antihydrogen review (Charlton et al, 2020) that discusses their impact on antihydrogen (line 105-106).

2- The literature presenting the constrains on ∆g/g from WEP-c experiments and the mentioned to be 'proposed positronium experiments' should also be cited accordingly. (line 115f)

--> We repeated here the citation for the positronium experiments in line 115 (from the introduction section). The WEP-c limits are claimed in multiple sources, here we cite the review from Charlton et al. 2020, that contains the further references.

3- In the abstract the concepts of the cold M source is claimed to enable improved M spectroscopy. Besides the mention of improvements through increased statistics and less systematics (line 27f) this topic is not further addressed. It would be great to e.g. slightly expand the M & WEP-c discussion in line 125ff. I.e. what is the current level, which are the mentioned next generation experiments and how would the M source presented in this manuscript be able to improve on this. (If even, coupling the necessary laser into the SFHe environment seems to be quite a challenge on first thoughts)

--> WEP-c experiments require measurements of atomic transitions (or ion trap frequencies) carried out in different gravitational potentials. The absolute sensitivity depends on what background potential do we take into account compared to a state far from all masses. This issue is tricky to handle, for further discussion see e.g. Savely Karschenboim’s paper (cited in the present manuscript, doi: 10.1134/S1063773709100028) that summarises well the necessary precision that is needed to be sensitive to possible changes if we take into account the Earth, or solar system, galactic, or larger structures as a basis of this potential. Existing mounium spectroscopy experiments were not yet evaluated on energy level changes on the scales usually discussed for antihydrogen or other atoms - namely, sidereal (yearly) variations in atomic frequencies while the Earth travels in the solar system’s potential. These latter measurements were the basis of the estimated WEP-c limits, 10^-6 on antihydrogen. In “next generation” experiments, namely the Mu-Mass experiment (see paper from P. Crivelli in the same SciPost issue) we may however reach precisions (and large enough datasets) that would be necessary to carry out WEP-c measurements.

4- The authors present and justify the use of SFHe very well in 31.3.1. However, when coming to estimate the muon-to-vacuum conversion (line 195f) are quick to neglect potential surface effects without much justification. Is this a learning from the experiments mentioned in line 178? Why cant M 'stick' to SFHe with some surface states similar to the 'conventional converters'?

--> Diluting atoms in superfluid helium goes rather differently than in other materials: (1) The large and positive chemical potential of M atoms in SFHe (against the vacuum state) ensures a negative work function of the same magnitude, which transforms to positive kinetic energy after the atoms are leaving the bulk. In general, SFHe is extremely inert and does not dilute with contaminants (with a few exceptions). (2) Unlike in other solids or liquids, diffusion of impurities gets faster as we cool the liquid, due to the absence of available thermal excitations. (Helium atoms themselves lose their identity in the quantum liquid, and participate in most interactions via these excitations too.) In other materials the decreasing temperature results in slower diffusion, and thermal energies are not sufficient anymore to supply the work function needed to remove atoms from the surface. (Similar behavior is well known with the chemically identical hydrogen - e.g. from removing H from metals, one needs to apply high temperatures.)

In case of a neutral impurity with low polarizabilities the bulk chemical potential is expected to be the dominant effect when crossing the liquid-vapour barrier. With charged impurities, most significantly with free electrons, weak image charge effects can result in shallow trapping potentials just above the surface, and create a 2D "sheet" of electrons in quantised states above the surface, outside of the bulk. For the existence of such significant trapping potentials the impurity has to have a net electric charge. It is known experimentally that e.g. thermalised hydrogen atoms are not sticking to the surface of superfluid helium, but bounce off elastically, even preserving polarisation.

What we meant by “surface effects” has to do actually with the waves on the surface. One thing we still need to study is the cross section of absorbing a ripplon (surface excitations of the superfluid helium) at these temperatures, or the effects of the actual bulk shape of the surface. We are actually studying alternative ways to omit these problems of the free surface, by constraining the meniscus.

5- In the presentation of the concept of the three-grating interferometer 31.3.2 many different quantities and values are quoted and justified in the text. It would be great to have them presented in a more concise way to conclude on the many aspects necessary to do this experiment, e.g. by introducing the values in Fig. 31.2.

--> The values in Fig. 31.2 are introduced from line 230 , and we added more clarification by fixing a label that was different on the figure and explaining each parameters.

6- The authors explain that the positioning of the grating G3 has to fulfill a certain translation accuracy (line 254f). While it is clearly linked, what about the accuracy needed in the angular alignment of the gratings? Moreover, even if much less stringent, the positioning of G1 and G2 will also have its constraints?

--> The sensitivity of the angular alignment is under study, hence it was not part of the present paper, but it is not expected to be the bottle neck of the work once the linear alignment and stabilisation issues are sorted out. The interferometer has the advantage that it is rather short (few centimeters), hence we expect to be able to fix the first two gratings G1 and G2 precisely compared to each other and stabilise it as a block, while carrying out the scans only with the third grating.

7- When the authors conclude on the signal rate and sensitivity of the proposed experiment most values used in the calculation are well justified. However, the detection efficiency is simply assumed to be 30% without any explanation (line 258). As this is a very critical parameter and the detection in such an environment is anything but easy the reviewer would appreciate if more details on how this value is derived would be given.

--> The above number is a pessimistic estimate, approximately half of what we could already achieve with our preliminary detection scheme (neglecting geometric efficiency). We are however still working on preliminary studies (measurements and Monte Carlo simulations) to have a better knowledge about the details. We expect to have here a high (>90%) positron detection efficiency by surrounding the device with position sensitive detectors. For verifying M atoms exiting the interferometer, we however require the coincident detection of an atomic electron. For this purpose electrons are collected and accelerated by electric fields from the decay region towards a cryogenic detector. The efficiency and background rejection of the system is mostly driven by the performance of the atomic electron collection and detection scheme, and with our preliminary measurements it is close to 70%. For an exact number however further studies are needed, now we simply added a sentence to verify the origin of this number.

8- The reviewer would like to ask the authors to also use the abbreviation for HIMB (line 268) and provide an adequate citation. Moreover, an increased rate almost always lead to an increased sensitivity. If possible, could the authors state a quantitative estimation?

--> We explained the abbreviation and added a citation. As mentioned in the paper, optimistically HIMB would add x100 in rate (which appears in the sensitivity equation as a 1/sqrt(N) factor), but this would be at a cost of a worse phase space, the details of which are still under discussion. Source was added in form of a link to the project.

9- The implementation of hyperlinks in the references should be done for all items. The reviewer would also like to ask the authors to perform this practice prior to their future submissions as it enables a reviewer to conduct a better review through spending more time on what matters.

--> We added the missing hyperlinks to the references.

Anonymous on 2021-08-12  [id 1659]

(in reply to Anna Soter on 2021-07-31 [id 1628])
Category:
answer to question

Many thanks for this throughly detailed reply. Much appreciated. It was a pleasure reviewing the paper! All the best.

---

## Round 1 · Referee Report · Adrian Signer (Referee 2) · 2021-7-5

Report

We (the editors Cy Hoffman, Klaus Kirch, Adrian Signer) had the
opportunity to review an earlier draft of the article and were in
communication with the authors before the submission. All our comments
and suggestions have been taken into account. Hence, we think the
paper can now be published in the current form.

---

## Round 2 · Author Response

Minor revisions requested by Referee 2 were addressed.

---

## Round 2 · List of Changes

Minor revisions requested by Referee 2 were addressed, see details and list of changes in Reply to Referee.

---

## Editorial Decision

published